# Incidence of the Acute Symptom of Chronic Periodontal Disease in Patients Undergoing Supportive Periodontal Therapy: A 5-Year Study Evaluating Climate Variables

**DOI:** 10.3390/ijerph16173070

**Published:** 2019-08-23

**Authors:** Hikari Saho, Noriko Takeuchi, Daisuke Ekuni, Manabu Morita

**Affiliations:** Department of Preventive Dentistry, Okayama University Graduate School of Medicine, Dentistry and Pharmaceutical Sciences, 2-5-1 Shikata-cho, Kita-ku, Okayama 700-8558, Japan

**Keywords:** acute-phase reaction, periodontal disease, climate, air pressure, temperature

## Abstract

Although patients under supportive periodontal therapy (SPT) have a stable periodontal condition, the acute symptom of chronic periodontal disease occasionally occurs without a clear reason. Therefore, in the present study, to obtain a better understanding of this relationship in patients undergoing SPT, we hypothesized that the acute symptom of chronic periodontal disease might be affected by climate factors. We conducted a questionnaire study and carried out oral examinations on patients undergoing SPT who had been diagnosed as having the acute symptom of chronic periodontal disease. We collected climate data from the local climate office in Okayama city, Japan. We predicted parameters that affect the acute symptom of chronic periodontal disease with unidentified cause and divided patients into high and low groups in terms of climate predictors. Then we defined the cut-off values of parameters showing significant differences in the incidence of the acute symptom of chronic periodontal disease. The incidence of the acute symptom of chronic periodontal disease with unidentified cause was significantly different when the cases were classified according to the maximum hourly decrease in barometric pressure (1.5 and 1.9 hPa) (*p* = 0.04 and *p* = 0.03, respectively). This suggests that climate variables could be predictors of the acute symptom of chronic periodontal disease. Therefore, gaining a better understanding of these factors could help periodontal patients undergoing SPT prepare to avoid the acute symptom of chronic periodontal disease.

## 1. Introduction

Periodontal disease is a chronic inflammatory disease initiated by pathogenic bacteria. It destroys the connective tissues or the bone that support teeth and induces periodontal pocket formation or gingival bleeding [1,2]. The host response to these microorganisms and their products is responsible for most of the destruction of periodontal tissue [3,4]. The aim of active periodontal treatment is to reduce the inflammatory response by removing bacterial deposits. First, basic periodontal treatment (plaque control, scaling, root planing, elimination of plaque retention factors, occlusal adjustment, and temporary splint) is given to all periodontal patients. After basic periodontal treatment, periodontal surgery is considered if active pockets with ≥4 mm depth remain. After periodontal surgery, oral rehabilitation such as repair and prosthetic procedures is performed to restore oral function [5]. After treatment and inflammation arrest, supportive periodontal therapy (SPT) is often applied to reduce the likelihood of reinfection and disease progression, help patients maintain teeth without pain, control excessive mobility or infection over the long term, and prevent other related oral diseases [6].

SPT is considered an essential part of overall periodontal control and important for the long-term stability of patients who have received successful treatment for periodontal disease [7,8,9]. Although the periodontal condition of patients undergoing SPT tends to be stable, acute symptoms, e.g., periodontal abscesses, occasionally occur [10]. A previous study reported the existence of intra-oral triggering factors, including a cracked tooth, local factors affecting root morphology, foreign bodies such as a piece of a toothpick, and the infection of lateral cysts [11]. However, the acute symptom of chronic periodontal disease occasionally occurs in some patients who show no such intra-oral factors.

Climate variables, such as maximum daily wind speed, maximum hourly range of temperature, and the maximum hourly range of barometric pressure, have been reported to be associated with a variety of diseases and symptoms, including blood pressure [12,13,14], cardiovascular disease [15,16], heatstroke [17], stroke [18], pediatric trauma [19], mortality [20], asthma [21,22], depression [23,24], rheumatoid arthritis [25,26], and pain in general [27]. Associations with climate variables have also been reported in the dental field for conditions such as temporomandibular disorders [28] and oral pain [29].

Our previous short-term pilot study found that climate variables significantly affected the incidence of the acute symptom of chronic periodontal disease [30]. Therefore, we hypothesized that climate factors might also have some effect on the incidence of the acute symptom of chronic periodontal disease over the long term [30]. To provide a stable basis for a proper time-series study over the long term, a study period of 5 years or more with at least 50 observations is needed [31]. The aim of the present study was to identify and establish possible cut-off values for climate conditions that might affect the incidence of the acute symptom of chronic periodontal disease and their relationship over a 5-year period.

## 2. Materials and Methods

### 2.1. Study Population

Data were collected from patients who had undergone SPT at the Preventive Dentistry Clinic of Okayama University Hospital in Okayama, Japan from November 2011 to October 2016. In our clinic, patients received SPT consisting of oral examinations (periodontal status assessed with bleeding on probing, probing pocket depth, clinical attachment level, oral hygiene conditions, occlusion, furcation involvement, and plaque retention factors), oral hygiene instructions, supra/sub-gingival debridement and scaling, and root planing every 3–4 months [5,32]. The inclusion criteria were patients who had clinical symptoms such as redness, swelling, pain, and a feeling of warmth in the periodontal lesion and received a diagnosis of acute symptom of chronic periodontal disease at a regularly scheduled SPT visit or an emergency visit during the SPT phase. [33,34]. The exclusion criteria were hospitalized patients, those with cancer, those with missing data, and those who were living outside of Okayama’s local climate area. All patients selected for inclusion provided written informed consent to participate in the study, which was conducted in accordance with the Declaration of Helsinki. The study protocol was approved by the Okayama University Hospital Ethics Committees (Nos. 502, 692).

### 2.2. Oral Examination

The oral status of the patients was assessed by qualified dentists. X-rays and/or oral examinations were used to distinguish the acute symptom of chronic periodontal disease from other diseases, such as apical periodontal disease, trauma, and tooth fracture.

### 2.3. Questionnaires

In order to confirm a history of the acute symptom of chronic periodontal disease, we collected the following information: (1) Date, (2) location, (3) possible trigger that the patients were aware of, and (4) the other perceptible trigger, such as physical and/or mental stress, occlusal trauma, and insufficient of oral hygiene. We defined the acute symptom of chronic periodontal disease as cases with unidentified cause in which no possible triggers could be clearly identified [30].

### 2.4. Climate Data

We obtained the following climate data from the local climate office in Okayama city: Daily humidity range; minimum daily humidity (%); maximum daily humidity (%); mean daily humidity (%); maximum hourly decrease in temperature (°C); maximum hourly increase in temperature (°C); daily temperature range (°C); minimum daily temperature (°C); maximum daily temperature (°C); mean daily temperature (°C); total daily rainfall (mm); total hours of sunlight (h); maximum hourly decrease in barometric pressure (hPa); maximum hourly increase in barometric pressure (hPa); minimum daily barometric pressure (hPa); maximum daily barometric pressure (hPa); mean daily barometric pressure (hPa); maximum daily humidity (%); minimum daily wind speed (m/s); maximum daily wind speed (m/s); and mean daily wind speed (m/s) [30,35,36].

### 2.5. Statistical Analysis

We investigated the association between these climate variables and the incidence of the acute symptom of chronic periodontal disease using the Box–Jenkins multivariate autoregressive integrated moving average (ARIMA) model [37]. We established models by determining the ARIMA model orders (p, d, q) using autocorrelation and partial autocorrelation, and then the model parameters using the unconditional least squares method. We then evaluated the model fits and statistical significance of the parameters [35], and based on the results, we predicted the parameters that would affect the acute symptom of chronic periodontal disease with unidentified cause.

Furthermore, we divided cases identified as the acute symptom of chronic periodontal disease with unidentified cause into high and low groups in terms of climate predictors, and then defined cut-off values for the parameters showing significant differences in the incidence of cases. We used the chi-squared test to compare the daily incidence between the high and low groups. The Statistical Package for the Social Sciences for Windows (version 21.0J; SPSS Japan, Tokyo, Japan) was used for all analyses, and values of *p* < 0.05 were considered statistically significant.

## 3. Results

Table 1 shows the characteristics of the total cohort of 47,859 patients. 674 (1.41%) of them had received a diagnosis of being in the acute symptom of chronic periodontal disease. Among these patients, 458 were in the acute symptom of chronic periodontal disease with unidentified cause. The mean age ± standard deviation was 68.2 ± 10.9 years.

Table 2 shows the correlation between the incidence of the acute symptom of chronic periodontal disease with unidentified cause and climate parameters. Although four climate parameters were correlated with the incidence of the acute symptom of chronic periodontal disease, only the association with mean daily temperature was significant (*p* = 0.027). According to the ARIMA model, the climate predictors of the acute symptom of chronic periodontal disease with unidentified cause were mean daily barometric pressure, minimum daily barometric pressure, maximum hourly decrease of barometric pressure, and mean daily temperature. The lag time was within 1–3 days after changes in the climate variables.

Figure 1 shows the monthly average incidence rate of the acute symptom of chronic periodontal disease with unidentified cause. The incidence rate was the highest in January (1.17%) and the lowest in May (0.82%). The ratio appeared to have an upward trend and to peak in the winter.

Figure 2 shows the cut-off values and the incidence rate of the maximum hourly decrease in barometric pressure. Based on the cut-off values, we classified the cases with unidentified cause into high and low incidence groups. The cut-off values for the maximum hourly decrease in barometric pressure (1.5 and 1.9 hPa) showed the following significant differences with regard to the incidence of cases. The incidence of the acute symptom of chronic periodontal disease with unidentified cause was 1.3% at 2 days after the day with ≥1.5 hPa maximum hourly decrease in barometric pressure, and 1.0% at 2 days with <1.5 hPa. A significant difference was observed between the two groups (*p* = 0.04). The incidence rate of the acute symptom of chronic periodontal disease with unidentified cause was 1.5% at 2 days after the day with ≥1.9 hPa maximum hourly decrease in barometric pressure and 0.9% at 2 days with <1.9 hPa. This difference was also significant between the two groups (*p* = 0.03).

## 4. Discussion

To the best of our knowledge, this is the first long-term study to investigate the association between the incidence of the acute symptom of chronic periodontal disease with unidentified cause and climate parameters using the ARIMA model. The results revealed that mean daily barometric pressure, minimum daily barometric pressure, maximum hourly decrease of barometric pressure, and mean daily temperature might be predictors of the acute symptom of chronic periodontal disease. In addition, we suggested cut-off values for climate factors. As periodontal disease is a highly prevalent disease, we consider it is meaningful that some predictors of its acute symptom may have been clarified. These predictors could serve as a useful tool for patients undergoing SPT to help avoid the acute symptom of chronic periodontal disease. For example, it may be possible to prevent the acute symptom of chronic periodontal disease by prior antibiotic treatment or forecasting.

The incidence rate of the acute symptom of chronic periodontal disease with unidentified cause was significantly different when we classified the cases into two groups based on the maximum hourly decrease in barometric pressure (1.5 and 1.9 hPa). The central pressure of an explosive cyclone decreases by about 1 hPa/h [38]. An hourly decrease in barometric pressure of 1.5 hPa or 1.9 hPa is relatively large, and such a decrease does not occur frequently (136 days and 50 days every 5 years, respectively). A relatively low incidence of these climate changes might trigger the events in which acute symptom occasionally occurs even in a long-term stable stage of chronic periodontal disease.

An animal study indicated that decreases in barometric pressure (5 hPa/h) result in pain and increases in blood pressure and heart rate [39]. It is considered that decreased barometric pressure causes tension to the sympathetic nervous system and increases secretion of adrenocortical hormones such as epinephrine [40]. These changes in hormone secretion and excitation of sympathetic nerves cause ischemia by contracting peripheral blood vessels and lowering oxygen concentration and pH in tissues; these conditions increase local pain activity [40]. Another previous study reported that decreasing barometric pressure affects some stress hormones, including catecholamines, such as epinephrine, norepinephrine, dopamine, and cortisol [41], which directly affect the growth of periodontal disease-related bacteria in vitro [42]. Therefore, decreases in barometric pressure might indirectly contribute to the incidence of the acute symptom of chronic periodontal disease. A case report also showed that a rapid decrease in barometric pressure, for example, on an airplane, can affect the status of acute apical periodontal disease [43]. Other studies have found that barometric pressure can affect a variety of symptoms, including myopia [44], passenger discomfort on aircrafts [45], sleep-disordered breathing [46], deep venous thrombosis [47], and oral pain [29,48]. Although a mechanism may underlie the relationship between barometric pressure and the incidence of the acute symptom of chronic periodontal disease remains difficult to explain [41].

In the present study, a relationship was also found between mean daily temperature and the incidence of the acute symptom of chronic periodontal disease. The incidence of the acute symptom of chronic periodontal disease with unidentified cause was 0.89% at 1 day after days with a mean temperature of ≥13 °C and 1.05% at 1 day after days with a mean daily temperature of <13 °C (*p* = 0.03) (Figure 3). In addition, it was 0.86% at 1 day after days with a mean daily temperature of ≥17 °C and 1.07% at 1 day after days with a mean temperature of <17 °C. A significant difference was observed between these two groups (*p* = 0.03) (Figure 3). A mean daily temperature of <17 °C is common in Okayama during the winter season, which supports the upward trend and peak incidence of the acute symptom of chronic periodontal disease in winter. A past study indicated that changes in temperature affect blood biomarkers. For instance, C-reactive protein and interleukin-6 levels increase with a 10 °C decrease in temperature [49]. Moreover, fibrinogen and plasminogen activator inhibitor levels have been found to increase in association with a 5 °C decrease in temperature in patients with a specific genetic background [50]. According to a previous study involving individuals with asthma and chronic obstructive pulmonary disease, cold air resulting from low temperatures can stimulate bioactive chemical release from granulocytes or mast cell degranulation and increase macrophage recruitment in the lower airways, which can lead to wheezing [51,52]. Another previous study reported that exposure to cold air reduces immunity to respiratory infections [53]. Low temperatures have also been reported to result in similar respiratory diseases in a number of different countries [54,55,56,57]. Therefore, the inflammatory process might be affected, at least in part, by temperature-related changes in blood biomarkers, even in periodontal tissue; however, the underlying mechanism remains unclear.

Among the sample in the present study, the occurrence of the acute symptom of chronic periodontal disease appeared within 1–3 days after climate episodes. This finding suggests a response time lag. During the initial inflammatory process in the acute symptom of chronic periodontal disease, pathogenic bacteria attack soft tissues surrounding the periodontal pocket [58]. These bacterial pathogens attract inflammatory cells to induce chemokines or cytokines and adjust the inflammatory response within 2–72 h [59,60,61,62]. This result was consistent with our previous pilot study and demonstrated good reproducibility.

In the present study, the incidence of the acute symptom of chronic periodontal disease in all patients undergoing SPT was 1.41%. Previous studies have reported that the prevalence of periodontal abscesses was 1.04–27.5% [63,64,65,66,67]. The prevalence of periodontal abscesses in the present study was also within this range. Therefore, our findings are not limited to a specific group and may be generalizable to other populations.

This study has some limitations. First, we obtained climate data from only one region. It is necessary to conduct a multicenter investigation including other geographical regions in the future. Second, further studies to visualize in vivo changes are required to explore the exact mechanisms underlying climate variability. Further research is needed to gain a better understanding of how climate variables interact with each other, and to elucidate the mechanism by which these factors relate to the incidence of the acute symptom of chronic periodontal disease. Third, we did not investigate timely analyses about systematic diseases at the incidence of the acute symptom of chronic periodontal disease. The chronic pathologies might have affected the incidence of the acute symptom of chronic periodontal disease [68,69]. Finally, we could not use disinfectants containing 0.12–0.50% of chlorhexidine [70] during SPT phase because the upper limit concentration is defined as 0.05% by the Japanese pharmaceutical affairs law. However, it is possible that some materials other than chlorhexidine may have suppressed the acute symptom of chronic periodontal disease.

## 5. Conclusions

The incidence of the acute symptom of chronic periodontal disease with unidentified cause is likely to occur within three days of a climate episode, such as changes in barometric pressure or temperature. In addition, the incidence of the acute symptom of chronic periodontal disease with unidentified cause significantly differed when cases were classified based on the maximum hourly decrease in barometric pressure (1.5 and 1.9 hPa).

## Figures and Tables

**Figure 1 ijerph-16-03070-f001:**
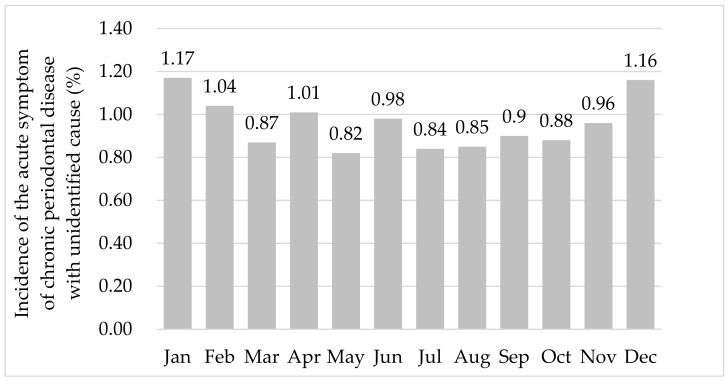
Monthly average ratio of the incidence of the acute symptom of chronic periodontal disease with unidentified cause.

**Figure 2 ijerph-16-03070-f002:**
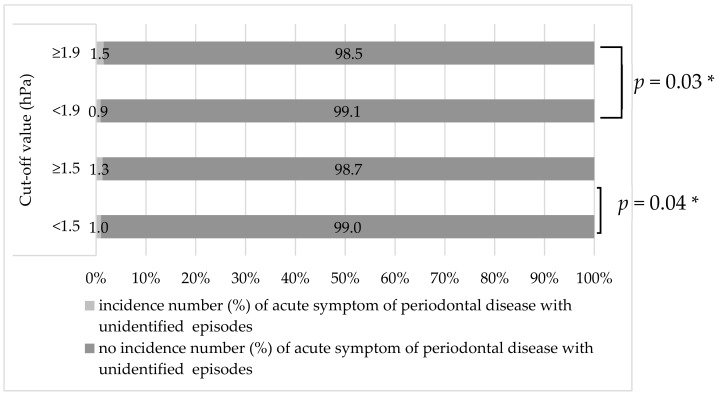
Evaluation of the two cut-off values of the maximum hourly decrease in barometric pressure to predict the incidence of the acute symptom of chronic periodontal disease. * Chi-squared test.

**Figure 3 ijerph-16-03070-f003:**
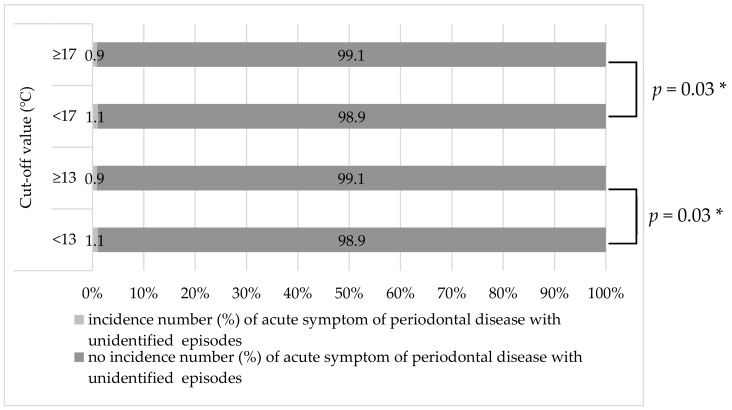
Evaluation of the two cut-off values of the mean daily temperature to predict the incidence of the acute symptom of chronic periodontal disease. * Chi-squared test.

**Table 1 ijerph-16-03070-t001:** Characteristics of the study population.

Age (Years)	Male	Female	Total
**Number of Patients Who Had Received Supportive Periodontal Therapy**			
20–29	198	327	525
30–39	294	932	1226
40–49	897	2170	3067
50–59	1623	4856	6479
60–69	4657	11,160	15,817
70–79	4685	10,047	14,732
≥80	2348	3665	6013
Total	14,702	33,157	47,859
**Number of Patients with Acute Symptom of Chronic Periodontal Disease with Unidentified Cause**			
20–29	1	2	3
30–39	1	4	5
40–49	9	16	25
50–59	19	26	45
60–69	46	103	149
70–79	62	116	178
≥80	16	37	53
Total	154	304	458

**Table 2 ijerph-16-03070-t002:** Climate predictors of the acute symptom of chronic periodontal disease with unidentified cause.

Parameter	Mean ± SD	Lag Time ^a^	*r*	*p* Value *
**Wind speed (m/s)**				
Mean daily wind speed	3.0 ± 1.2		Omitted	
Maximum daily wind speed	6.6 ± 2.5		Omitted	
Minimum daily wind speed	0.6 ± 0.6		Omitted	
**Barometric Pressure (hPa)**				
Mean daily barometric pressure	1013.5 ± 7.0	3	0.242	0.896
Maximum daily barometric pressure	1016.3 ± 7.0		Omitted	
Minimum daily barometric pressure	1010.7 ± 7.2	3	0.260	0.884
Daily range of barometric pressure	5.5 ± 3.3		Omitted	
Maximum hourly increase in barometric pressure	0.8 ± 0.4		Omitted	
Maximum hourly decrease in barometric pressure	1.0 ± 0.4	2	0.043	0.123
Total hours of sunlight (h)	13.7 ± 9.6		Omitted	
Total daily rainfall (mm)	3.5 ± 10.0		Omitted	
**Temperature (°C)**				
Mean daily temperature	16.3 ± 8.6	1	0.009	0.027
Maximum daily temperature	20.7 ± 8.8		Omitted	
Minimum daily temperature	12.3 ± 8.9		Omitted	
Daily range of temperature	8.4 ± 2.9		Omitted	
Maximum hourly increase in temperature	2.1 ± 0.8		Omitted	
Maximum hourly decrease in temperature	1.5 ± 0.7		Omitted	
**Humidity (%)**				
Mean daily humidity	67.0 ± 11.4		Omitted	
Maximum daily humidity	85.5 ± 8.7		Omitted	
Minimum daily humidity	46.1 ± 14.5		Omitted	
Daily humidity range	39.4 ± 11.5		Omitted	

^a^ Represents the delay necessary to observe the effect (in days). * ARIMA model.

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
