# Peer review of "Incidence of the Acute Symptom of Chronic Periodontal Disease in Patients Undergoing Supportive Periodontal Therapy: A 5-Year Study Evaluating Climate Variables"

_ijerph, 2019, doi:10.3390/ijerph16173070_

Round 1
Reviewer 1 Report
Thank you for submitting this very interesting and thought-provoking review.
Please consider the following for revision.
You use the term ‘acute symptom of chronic periodontitis’ or ‘periodontal disease in the acute symptom’ liberally throughout the paper. I am not familiar with these diagnoses. I am assuming you are using the 1999 (Armitage) classification. Are you referring to ‘Periodontal abscess’ or acute loss of probing/clinical attachment levels? Please clarify and use accepted terminology.
Please have someone with English native speaker/writing skills review for language uses. For example line 32 – ‘demolishes’ is not an appropriate verb here. Line 42 – beginning the sentence with two conjunctions. etc.
Line 67 reference [31]. Please cite a more appropriate reference for the SPT procedure.
Line 70 reference [32] Please explain how this reference is appropriate here.
Lines 66 to 70 are confusing. Are periodontal probing depths and bleeding sites assessed at an SPT visit? Did these patients present with acute symptoms? Were patients undergoing SPT because of symptoms, or were they regularly schedule for SPT? Were the findings incidental to a regularly scheduled SPT visit? How is ‘warmth’ assessed?
Line 76. What is a ‘qualified’ dentist? How did assessments? Were they calibrated? Or did they simply rule out ‘apical periodontal disease, trauma, tooth fracture’
Line 82. Are ‘physical and mental stress’ part of acute symptoms? (or does that belong to trigger?)
Line 95. Please cite a reference for the Okoyama city office for weather statistics.
Line 99 Please cite a primary reference for the ARIMA model, or give some background why you are referencing [34, 35]
Lines 175 to 181. Data not shown for temperature? Please show it in a table or figure.
Thank you for considering these. I am looking forward to your revisions.
Reviewer 2 Report
Dear Authors, This study is very interesting and it is a very current topic.
I suggest a major revision:
In keyword section please use MeSH words (https://meshb.nlm.nih.gov/search)
In Introduction section please cite some information about periodontitis course and therapy
in order to limit the risk of error it would be the case to mention, in the "discussion" section, that some chronic pathologies can favor the onset of periodontal disease. "for completeness of the study, it is necessary to report that some chronic diseases could alter some results, since the latter have important correlations with periodontal disease. Among these pathologies we certainly recognize the diabetic disease and the celiac disease." (Cervino, G.; Fiorillo, L.; Laino, L.; Herford, A. S.; Lauritano, F.; Giudice, G. L.; Fama, F.; Santoro, R.; Troiano, G.; Iannello, G.; Cicciu, M., Oral Health Impact Profile in Celiac Patients: Analysis of Recent Findings in a Literature Review. Gastroenterol Res Pract 2018, 2018, 7848735. -----Cervino, G.; Terranova, A.; Briguglio, F.; De Stefano, R.; Famà , F.; D'Amico, C.; Amoroso, G.; Marino, S.; Gorassini, F.; Mastroieni, R.; Scoglio, C.; Catalano, F.; Lauritano, F.; Matarese, M.; Giudice, R. L.; Siniscalchi, E. N.; Fiorillo, L., Diabetes: Oral health related quality of life and oral alterations. BioMed Research International 2019, 2019.)
Some innovative materials have recently shown effective results in topical disinfection of the oral cavity, with important results in periodontal and non-surgical periodontal phases. Such as chlorhexidine. Please cite them (https://doi.org/10.3390/gels5020031)
Thank You
Round 2
Reviewer 2 Report
Manuscript has been improved and is now suitable for publication in present form